# Improved VMD-ELM Algorithm for MEMS Gyroscope of Temperature Compensation Model Based on CNN-LSTM and PSO-SVM

**DOI:** 10.3390/mi13122056

**Published:** 2022-11-24

**Authors:** Xinwang Wang, Huiliang Cao

**Affiliations:** 1School of Instrument Science and Engineering, Southeast University, Nanjing 210018, China; 2Key Laboratory of Instrumentation Science & Dynamic Measurement, Ministry of Education, North University of China, Taiyuan 030051, China

**Keywords:** MEMS gyroscope, convolutional neural networks—long short-term memory (CNN-LSTM), particle swarm optimization—support vector machines (PSO-SVM), variational modal decomposition (VMD), temperature compensation

## Abstract

The micro-electro-mechanical system (MEMS) gyroscope is a micro-mechanical gyroscope with low cost, small volume, and good reliability. The working principle of the MEMS gyroscope, which is achieved through Coriolis, is different from traditional gyroscopes. The MEMS gyroscope has been widely used in the fields of micro-inertia navigation systems, military, automotive, consumer electronics, mobile applications, robots, industrial, medical, and other fields in micro-inertia navigation systems because of its advantages of small volume, good performance, and low price. The material characteristics of the MEMS gyroscope is very significant for its data output, and the temperature determines its accuracy and limits its further application. In order to eliminate the effect of temperature, the MEMS gyroscope needs to be compensated to improve its accuracy. This study proposed an improved variational modal decomposition—extreme learning machine (VMD-ELM) algorithm based on convolutional neural networks—long short-term memory (CNN-LSTM) and particle swarm optimization—support vector machines (PSO-SVM). By establishing a temperature compensation model, the gyro temperature output signal is optimized and reconstructed, and the gyro output signal with better accuracy is obtained. The VMD algorithm separates the gyro output signal and divides the gyro output signal into low-frequency signals, mid-frequency signals, and high-frequency signals according to the different signal frequencies. Once again, the PSO-SVM model is constructed by the mid-frequency temperature signal to find the temperature error. Finally, the signal is reconstructed through the ELM neural network algorithm, and then, the gyro output signal after noise is obtained. Experimental results show that, by using the improved method, the output of the MEMS gyroscope ranging from −40 to 60 °C reduced, and the temperature drift dramatically declined. For example, the factor of quantization noise (Q) reduced from 1.2419 × 10^−4^ to 1.0533 × 10^−6^, the factor of bias instability (B) reduced from 0.0087 to 1.8772 × 10^−4^, and the factor of random walk of angular velocity (N) reduced from 2.0978 × 10^−5^ to 1.4985 × 10^−6^. Furthermore, the output of the MEMS gyroscope ranging from 60 to −40 °C reduced. The factor of Q reduced from 2.9808 × 10^−4^ to 2.4430 × 10^−6^, the factor of B reduced from 0.0145 to 7.2426 × 10^−4^, and the factor of N reduced from 4.5072 × 10^−5^ to 1.0523 × 10^−5^. The improved algorithm can be adopted to denoise the output signal of the MEMS gyroscope to improve its accuracy.

## 1. Introduction

The micro-electro-mechanical system (MEMS) gyroscope has become a promising sensing technology to arouse research interests due to its advantages of small size, high accuracy, and long durability, and it is widely used in the fields of inertial navigation and positioning. Meanwhile, it has disadvantages including long initial installation [1] and calibration time [2], more temperature-sensitive equipment and materials [3], and accumulated angle integration errors over time. Specifically, due to the difference in the expansion coefficient of the internal construction materials of the MEMS gyroscope, corresponding thermal resistance and thermal stress will be generated [4], and the change of the environment temperature has a greater impact on the stability of the MEMS gyroscope’s bias. To address these problems, the temperature drift model is usually established to suppress the temperature drift of the gyroscope and improve the accuracy of the MEMS gyroscope [5,6], the relationship between the output of the gyroscope and the temperature drift is predicted and compensated by analyzing the model, and then, the internal structure of the MEMS gyroscope is optimized to suppress the temperature drift [5].

Until now, some attempts have been made to improve the above deficiencies by studying temperature characteristics of the structure and peripheral circuit of MEMS gyroscopes [6]. For example, Nesterenko et.al. [7] presents the design and simulation of a microelectromechanical gyroscope that simultaneously determines two components of angular velocity. In order to consider how temperature influences eigenfrequencies and informative vibrational magnitude of the micromechanical angular velocity sensor. Guo et al. [8] adopted the finite element method (FEM) to verify the feasibility of the design and to compute the performance of the system. Results show that in the range of −20 to 80 °C, the maximum relative error of the resonant frequency variation reducing from 16.3% to 3.1% indicates that this design scheme is effective in overcoming the temperature effect of this kind of gyroscope. Fu et al. [9] designed a constant trans-conductance high-linearity amplifier that could realize the low-phase drift and low-amplitude drift interface circuit at all temperatures, and the test result shows that the zero-point drift is lower than 30°/h (1-sigma) at the temperature range from −40 to 60 °C after three-order compensation made by the driving force. Cao et al. [10] presents the bandwidth-expanding method with a wide temperature range for sense mode coupling with a dual-mass MEMS gyroscope, and the turntable test results show that the sensing closed loop works stably in a wide temperature range (from 40 to 60 °C), and the bandwidth values are 107 and 97 Hz. Cao et al. [11] also proposed a sense mode closed-loop method for a dual-mass MEMS gyroscope based on the bipole temperature compensation method, and the bias temperature coefficient decreased from 9.534° h/°C to 5.991° h/°C. Cao et al. [12] demonstrated a closed-loop controlling system for the MEMS gyroscope sense mode, which can reduce bias temperature coefficients from 10.59° h/°C to 3.59° h/°C. Wang et al. [13] developed a digital output disk resonator gyroscope (DRG) on-chip temperature compensation method by using the virtual temperature sensor to complete the on-chip temperature compensation of the DRG angular velocity output, and the second-order compensation realizes the scale factor change of 40 ppm/°C and zero-output change of 27°/h over the full temperature range varying from −40 to 60 °C.

The above research proved that the accuracy of the MEMS gyroscope can be improved by improving the peripheral hardware circuit of the MEMS gyroscope’s structure; however, this type of method has the disadvantages of a long research period, cumbersome research content, and unstable results, which have an adverse effect on the accuracy of the MEMS gyroscope. Therefore, many scholars have devoted to improving the above deficiencies by studying temperature modal of MEMS gyroscopes. Shen et al. [14] proposed a novel multiple inputs/single output model based on the genetic algorithm (GA) and Elman neural network (Elman NN), and the comparison results between the traditional temperature-based model and the proposed multi-input/single output model show that the Allan variance coefficients are decreased, specifically, quantization noise (Q) from 0.0012 to 8.37 × 10^−4^, N from 1.19 × 10^−5^ to 5.01 × 10^−6^, bias instability (B) from 2.69 × 10^−4^ to 1.24 × 10^−4^, K from 0.0013 to 5.91 × 10^−4^, R from 9.48 × 10^−4^ to 3.27 × 10^−4^; the comparison results between the GA–Elman NN and traditional Elman NN show that the modeling accuracy is effectively improved, specifically in Allan variance coefficients, Q from 0.0051 to 8.37 × 10^−4^, random walk of angular velocity (N) from 3.08 × 10^−5^ to 5.01 × 10^−6^, B from 8.02 × 10^−4^ to 1.24 × 10^−4^. Wang et al. [15] introduced a new method that includes the radial basis function neural network (RBF NN), RBF NN based on the genetic algorithm (GA), and RBF NN based on GA with Kalman filter (KF). The experimental results proved the correctness of these three methods, and the MEMS gyroscope temperature energy influence drift is compensated effectively. Shen et al. [16] also developed a temperature error processing method for a dual-mass MEMS gyroscope based on a multi-scale parallel model. Compared with the conventional serial model, the proposed parallel model can eliminate the temperature errors more effectively with each parameter of Allan analysis improved. Specifically, the factor Q reduced from 0.035 to 9.93 × 10^−4^, the factor N reduced from 2.13 × 10^−5^ to 7.94 × 10^−6^, and B reduced from 5.28 × 10^−4^ to 4.79 × 10^−4^. Wu et al. [17] created an adaptive multi-scale method based on the combination of a generalized morphological filter (CGMF) for denoising of the output signal from a MEMS gyroscope. The proposed algorithm has a better performance in ARW and bias instability, which shows better advantages in gyroscope denoising. Shi et al. [18] described a double hidden layer long short-term memory (LSTM), which is presented to predict temperature data for the gyroscope (including single point and period prediction), and the LSTM network can be used to predict the temperature (time series data). Ma et al. [19] proposed a parallel denoising model based on PE-ITD and SA-ELM, and the result shows that the bias stability improves from 0.1874°/h to 1.599 × 10^−3^°/h at a temperature varying from −40 to 60 °C (enhanced by 99.1%), which indicated that the designed new method is more accurate and effective. In Chang’s [20] study, a denoising and temperature drift compensation parallel model method based on wavelet transform and forward linear prediction (WFLP) and support vector regression based on the cuckoo search algorithm (CS-SVR) was proposed, for the sake of decreasing the effects of noise and temperature error on the measurement accuracy of MEMS gyroscopes. Experimental results demonstrated that the proposed method can decrease noise and compensate for the temperature error effectively. Gu et al. [21] proposed a bias drift self-calibration method for MEMS gyroscopes based on noise-suppressed mode reversal without the modeling of a bias drift signal. The experimental results show that the proposed method is feasible and could achieve a better performance than the typical mode reversal. Song et al. [22] established a real-time wavelet denoising method used for the error compensation of MEMS gyroscopes, and the results show that the 1σ standard deviation of the residual signal is 0.0243°/s, and the 1σ standard deviation of the residual signal is 0.0175°/s after the noise reduction by the proposed method. Ding et al. [23] proposed an improved variational mode decomposition-wavelet threshold denoising (WTD) method to enhance the performance of MEMS gyroscopes. For the static signal, the mean square error (MSE) of the proposed method reduces by 10.1%, and the signal-to-noise ratio (SNR) increases by 14.2%. For the dynamic signal, the MSE of the proposed method decreases by 16.9%, and the SNR increases by 18.8%. Zhang et al. [24] proposed a serial-parallel estimation model (SPEM)-based sliding mode control (SMC) of MEMS gyroscopes. The simulation results show that the proposed controller obtains higher tracking accuracy and faster convergence, while the compound nonlinearity approximation has higher precision, and the proposed scheme is verified by simulations. Wang et al. [25] proposed a new model based on fusing an unscented Kalman filter (UKF) with support vector regression (SVR) optimized by the adaptive beetle antennae search (ABAS) algorithm. The experimental results show that the noise intensity (NI) and Durbin–Watson (DW) value of the proposed scheme in terms of the compensation accuracy for random drift data reduces and improves by 28.57% and 9.06%, respectively, compared with the conventional method. Huang et al. [26] proposed a calibration method based on deep learning concentrating on MEMS IMU gyroscopes. In this method, the output model of a MEMS IMU gyroscope is constructed by using the temporal convolutional network. The experimental results show that the attitude and position accuracy obtained by the inertial navigation solution using the compensated gyroscope data are improved compared with the existing MEMS sensor error compensation method based on deep learning, which proves that the proposed method can effectively and accurately calibrate the gyroscope error. Although there is much research on the output error of MEMS gyroscopes, temperature compensation models, and corresponding filtering algorithms, these algorithms still have many shortcomings in terms of computing speed and filtering capabilities. The temperature compensation proposed in this study is based on compensation temperature models and filtering algorithms to make up for this defect and improve the accuracy of the MEMS gyroscope.

In this study, a LSM6DS3 MEMS gyroscope with a novel structure and model was created. An improved VMD method based on convolutional neural networks—long short-term memory (CNN-LSTM) and particle swarm optimization—support vector machines (PSO-SVM) was proposed to deal with the temperature error. The output data of the X axis under the ranges of temperature from −40 to 60 °C were discussed. Furthermore, the proposed methods were compared with the Allan variance analysis method corresponding to the performance of the MEMS gyroscope to improve practicability and significance of the suggested methods.

## 2. Structure of MEMS Gyroscope

As is shown in Figure 1, the structure of this MEMS gyroscope is composed of an anchor, Coriolis mass, drive frame, sense frame, drive comb, sense comb sense spring, and drive spring. The structure of this MEMS gyroscope can be divided into two modes, namely drive mode and sense mode, which can be described as a spring–mass–damping second order system. The drive mode and the sense mode have one DOF along the X direction and Y direction, respectively. Each axis has two sections, which are stiffness and damping. As is described in Figure 1, Coriolis mass has two DOF on both the X axis and Y axis. In general, MEMS gyroscopes work on the drive mode by driving stiffness and drive damping, and the sense stiffness and sense damping are driven by the sensing mode to detect angular velocity when the Coriolis mass is subjected to an angular velocity of Z axis. The dynamic system can be represented by the simplified equations [27,28].
(1)$$m\ddot{x}+d_{x}\dot{x}+k_{x}x=F_{d}\sin(w_{d}t)$$
(2)$$m\ddot{y}+d_{y}\dot{y}+k_{y}y=-2m\Omega_{z}\dot{x}$$
where m is the main mass of this MEMS gyroscope, which is used by the drive mode and sense mode. $F_{d}$ is drive force amplitude, $w_{d}$ is drive force angular frequency, and $\Omega_{z}$ is angular rate around the Z axis.

By solving Equation (1), the displacement of the MEMS gyroscope in the drive direction can be described as:(3)$$\begin{array}{l} x(t)=\frac{2F_{d}\xi_{1}w_{n1}w_{d}/m_{x}}{{\left(w_{n1}^{2}-w_{d}^{2}\right)}^{2}+4\xi_{1}^{2}w_{n1}^{2}w_{d}^{2}}e^{-\xi_{1}w_{n1}t\cos(w_{n1}t\sqrt{1-\xi_{1}^{2}})}\\ +\frac{F_{d}w_{d}(2\xi_{1}^{2}w_{n1}^{2}+w_{d}^{2}-w_{n1}^{2})/m_{x}}{w_{n1}\sqrt{1-\xi_{1}^{2}}[{\left(w_{n1}^{2}-w_{d}^{2}\right)}^{2}+4\xi_{1}^{2}w_{n1}^{2}w_{d}^{2}]}e^{-\xi_{1}w_{n1}t\sin(w_{n1}t\sqrt{1-\xi_{1}^{2}})}\\ +\frac{F_{d}/m_{x}}{w_{n1}^{2}\sqrt{{\left(1-\frac{w_{d}^{2}}{w_{n1}^{2}}\right)}^{2}+{\left(2\xi_{1}\frac{w_{d}}{w_{n1}}\right)}^{2}}}\sin(w_{d}t-\varphi_{1}) \end{array}$$

By solving Equation (2), the displacement of the MEMS gyroscope in the sense direction can be described as:(4)$$\begin{array}{l} y(t)=\frac{2m_{y}\Omega_{Z}A_{1}w_{d}[2\xi_{2}w_{n2}w_{d}\sin\varphi_{1}+(w_{n2}^{2}-w_{d}^{2})\cos\varphi_{1}]/m_{y}}{{\left(w_{n2}^{2}-w_{d}^{2}\right)}^{2}+4\xi_{2}^{2}w_{n2}^{2}w_{d}^{2}}e^{-\xi_{2}w_{n2}t}\cos(w_{n2}t\sqrt{1-\xi_{2}^{2}})\\ -\frac{2m_{y}\Omega_{Z}A_{1}w_{d}[\xi_{2}w_{n2}(w_{n2}^{2}-3w_{d}^{2})\cos\varphi_{1}+w_{d}(2\xi_{2}^{2}w_{n2}^{2}+w_{n2}^{2}-w_{d}^{2})\sin\varphi_{1}]/m_{y}}{w_{n2}t\sqrt{1-\xi_{2}^{2}}[{\left(w_{n2}^{2}-w_{d}^{2}\right)}^{2}+4\xi_{2}^{2}w_{n2}^{2}w_{d}^{2}]}e^{-\xi_{2}w_{n2}t}\sin(w_{n2}t\sqrt{1-\xi_{2}^{2}})\\ +\frac{2m_{y}\Omega_{Z}A_{1}w_{d}/m_{y}}{w_{n2}^{2}\sqrt{{\left(1-\frac{w_{d}^{2}}{w_{n2}^{2}}\right)}^{2}+{\left(2\xi_{2}\frac{w_{d}}{w_{n2}}\right)}^{2}}}\cos(w_{d}t-\varphi_{1}-\varphi_{2}) \end{array}$$

Figure 2 illustrates that the circuit structure consists of the driving mode and sensing mode. The driving mode is composed of the MEMS gyroscope structure, amplifying phase, rectifier, low pass filter (LPF), DC voltage component, multiplier, and adder. The sense mode includes four different kinds of low pass filter (LPF), two different signal amplifiers, and two multipliers, which represent resonant frequency of driving mode and quality factor of driving mode, respectively. As for the drive mode, all driving mode loops form a self-excited oscillation loop so that the driving mode of the MEMS gyroscope always works in the resonance mode. As for the sense mode, this MEMS gyroscope adopts an open-loop sense mode; thus, the influence of the quadrature signals on circuit accuracy cannot be eliminated.

Figure 2 also displays the curve of the voltage of the MEMS gyroscope drive signal, sense signal and the quadrature signal along with time. The drive signal, sense signal, and quadrature signal are all varying by sine or cosine along with time. Due to the existence of the quadrature signal, the output error of the gyroscope is large, and the overall accuracy of the gyroscope is low. To address this problem, some software methods were proposed in this study to improve the accuracy of the MEMS gyroscope.

Figure 3 and Figure 4 show that the MEMS gyroscope has two corrected modes and four irrelevant modes. One of the corrected modes is the driving mode (Figure 3a) with a simulated resonance frequency of 11,034.6 Hz, and the other is the sensitive mode (Figure 3b) with a simulated resonance frequency of 11,035.2 Hz. In the driving mode, the Coriolis mass moves in resonance along the X-axis. In the sensitive mode, the same Coriolis mass moves in resonance along the Y axis because this structure is a fully decoupled structure, and the X-axis driving mode and the Y-axis sensitive mode do not interfere with each other. Figure 4 shows that the frequency difference between the driving mode and the sensitive mode is 0.6 Hz, and each mode is a fully decoupled mode independent of each other.

## 3. Algorithms

### 3.1. The Algorithm of VMD

The variational modal decomposition (VMD) algorithm was proposed by K. Dragomiretskiy, who argued that a complex signal is composed of sub-signals with different frequency bandwidths. The function of VMD is to decompose the composite signal into multiple sub-modes according to the frequency bandwidth. The VMD method completes the decomposition of the signal and the acquisition of signal components under the variational framework and decomposes the original signal adaptation by constructing and solving the variational constraint problem. In the application of fault signal processing, it can effectively decompose modal components with different center frequencies and bandwidths according to the frequency domain characteristics of the signal; meanwhile, it is not easily affected by frequency changes and has good noise robustness.

The first step is to construct the variational problem:

VMD is used to process fault signals and is mainly responsible for decomposing the signal into modal components with their respective optimal center frequencies and limited bandwidths according to the actual situation and given values. The objective function is the minimum sum of the bandwidths of the decomposed modal components.

The constrained variational problem model is as follows:(5)$$\{\begin{array}{c} \min\left\{\left\{u_{k}\right\},\left\{w_{k}\right\}\sum_{k}\delta (t)+\frac{j}{\pi t}*u_{k}(t)e^{-jw_{k}t_{2}^{2}}\right\}\\ s.t.\sum_{k}u_{k}=f \end{array}$$
where the K modal components are expressed as $\left\{u_{k}\right\}=\left\{u_{1},\ldots ,u_{k}\right\}$, the center frequency is expressed as $\left\{w_{k}\right\}=\left\{w_{1},\ldots ,w_{k}\right\}$, and δ(t) is the unit impulse function.

The second step is to solve the variational problem. After completing construction of the first step, the next step is to solve the above equation and to convert it into an unconstrained problem. Two parameters, quadratic penalty factor α and Lagrange multiplication operator λ(t), need to be introduced into the solution. The robustness of quadratic penalty factor α noise is strong, and it can ensure that the fault signal can be reconstructed well even if it is disturbed by the surrounding environment and other factors. The introduction of the Lagrange multiplication operator λ(t) can significantly change the constraints of the conditions of the variational problem and turn it into an unconstrained problem. The extended Lagrange function expression obtained by the combination of the two is:(6)$$\begin{array}{l} L(\left\{u_{k}\right\},\left\{w_{k}\right\},\lambda )=\alpha \sum_{k}\left\{\left[\delta (t)+\frac{j}{\pi t}*u_{k}(t)\right]\right\}e^{-jw_{2}t_{2}^{2}+f(t)-\sum_{k}u_{k}{\left(t\right)}_{2}^{2}}\\ +\left(\lambda (t),f(t)-\sum_{k}u_{k}(t)\right) \end{array}$$

The third step is to find the saddle point of the extended Lagrange function above, that is, the optimal solution of the expression of Equation (6). Using the alternating direction multiplier algorithm to solve the original minimization problem, it is necessary to continuously update $u_{k}^{n+1}$, $w_{k}^{n+1}$ and $\lambda^{n+1}$. The expression of $u_{k}^{n+1}$ is shown below:(7)$$u_{k}^{n+1}=\underset{u_{k}\in X}{\arg\min}\left\{\alpha {\left\|\partial_{t}\left[\left(\delta (t)+\frac{j}{\pi t}*u_{k}(t)\right)\right]e^{-jw_{k}t}\right\|}_{2}^{2}+{\left\|f(t)-\sum_{i}u_{i}(t)+\frac{\lambda (t)}{2}\right\|}_{2}^{2}\right\}$$

For simplicity, let $w_{k}=w_{k}^{n+1}$, then $\sum_{i}u_{i}(t)=\sum_{i\neq k}u_{i}{\left(t\right)}^{n+1}$, and Equation (8) is processed into the frequency domain space form by transformation as follows:(8)$${\hat{u}}_{k}^{n+1}=\underset{{\hat{u}}_{k},u_{k}\in X}{\arg\min}\left\{\begin{array}{l} \alpha {\left\|jw\left[\left(1+\mathrm{sgn}(w+w_{k})\right){\hat{u}}_{k}(w+w_{k})\right]\right\|}_{2}^{2}\\ +{\left\|\hat{f}(w)-\sum_{i}{\hat{u}}_{i}(w)+\frac{\hat{\lambda }(w)}{2}\right\|}_{2}^{2} \end{array}\right\}$$

w in the first term of the above Equation is expressed by $w-w_{k}$, and there are
(9)$${\hat{u}}_{k}^{n+1}=\underset{{\hat{u}}_{k},u_{k}\in X}{\arg\min}\left\{\begin{array}{l} \alpha {\left\|j(w-w_{k})\left[\left(1+\mathrm{sgn}(w)\right){\hat{u}}_{k}(w)\right]\right\|}_{2}^{2}\\ +{\left\|\hat{f}(w)-\sum_{i}{\hat{u}}_{i}(w)+\frac{\hat{\lambda }(w)}{2}\right\|}_{2}^{2} \end{array}\right\}$$

Using the conjugate symmetry of the signal, the non-negative frequency interval of Equation (9) is integrated as follows:(10)$${\hat{u}}_{k}^{n+1}=\underset{{\hat{u}}_{k},u_{k}\in X}{\arg\min}\left\{\int_{0}^{\infty }4\alpha {\left(w-w_{k}\right)}^{2}{\left|{\hat{u}}_{k}(w)\right|}^{2}+2\left|\hat{f}(w)-\sum_{i}{\hat{u}}_{i}(w)+\frac{\hat{\lambda }(w)}{2}\right|dw\right\}$$

The equation is quadratically optimized in the positive frequency range to obtain
(11)$${\hat{u}}_{k}^{n+1}=\frac{\hat{f}(w)-\sum_{i\neq k}{\hat{u}}_{k}^{}(w)+\frac{\hat{\lambda }(w)}{2}}{1+2\alpha {\left(w-w_{k}\right)}^{2}}$$
where ${\hat{u}}_{k}^{n+1}(w)$, $\hat{f}(w)$, and $\hat{\lambda }(w)$ are the Fourier transforms corresponding to $u_{k}^{n+1}$, f(w) and λ(w), respectively. ${\hat{u}}_{k}^{n+1}(w)$ is the Wiener filter of the current residual $\hat{f}(w)-\sum_{i\neq k}{\hat{u}}_{k}^{}(w)+\frac{\hat{\lambda }(w)}{2}$. In the same way, the center frequency $w_{k}^{n+1}$ can be expressed as:(12)$$w_{k}^{n+1}=\underset{w_{k}}{\arg\min}\left\{\partial_{t}\left[\left(\delta (t)+\frac{j}{\pi t}\right)*u_{k}(t)\right]e^{-jw_{k}t_{2}^{2}}\right\}$$

Equation (13) is obtained by converting to the frequency domain interval.
(13)$$w_{k}^{n+1}=\underset{w_{k}}{\arg\min}\left\{\int_{0}^{\infty }{\left(w-w_{k}\right)}^{2}{\left|{\hat{u}}_{k}(w)\right|}^{2}dw\right\}$$

The updated method of solving the center frequency is expressed in Equation (14).
(14)$$w_{k}^{n+1}=\frac{\int_{0}^{\infty }w{\left|{\hat{u}}_{k}(w)\right|}^{2}dw}{\int_{0}^{\infty }{\left|{\hat{u}}_{k}(w)\right|}^{2}dw}$$

The VMD algorithm decomposes the original signal through loop iteration and obtains the corresponding number of effective IMF components by presetting the value of K in the process of decomposition. On variational problem solving, the model is first transformed from the time domain to the frequency domain, the parameters $u_{k}$ and $w_{k}$ are continuously optimized and updated, and finally, the inverse Fourier transform is carried out to the time domain to obtain the center frequency of each mode.

The complete VMD algorithm follows these six steps with the detailed process illustrated in Figure 5.(1)Initialize $\left\{{\hat{u}}_{k}^{1}\right\},\left\{w_{k}^{1}\right\},{\hat{\lambda }}^{1}$, let n=0.(2)n=n+1, start the cycle.(3)Update $u_{k}$ according to Equations (3)–(17), update $w_{k}$ according to Equations (3)–(20).(4)k=k+1, repeat step (3). If k=K, the cycle ends, and if k≠K, the cycle continues.(5)λ is updated according to the following expression.
(15)$${\hat{\lambda }}^{n+1}(w)\leftarrow {\hat{\lambda }}^{n}(w)+\tau (\hat{f}(w)-\sum_{k}{\hat{u}}_{k}^{n+1}(w))$$(6)Steps (2) to (5) are repeated, with the given precision convergence criterion, and $\frac{\sum_{k}\left\|{\hat{u}}_{k}^{n+1}-{\hat{u}}_{k}^{n}\right\|}{\left\|{\hat{u}}_{k}^{n}\right\|}<\varepsilon$ is judged to whether it is satisfied. If so, turn to this step and stop the iteration to obtain IMF components. Otherwise, skip to step (2).

### 3.2. The Algorithm of CNN-LSTM

The advantages of both CNN and LSTM networks are combined to propose a novel algorithm. First of all, the CNN network performs well in extracting N-gram features at different positions in neurons, which can be extracted by convolution operations, so that it can be used to identify noise in gyro output signals. The LSTM network can deal with the noise of any length and extract the dependencies between the noise. After combining the two, it can benefit from the advantages of the two networks and identify different noise in the gyro output signal more accurately and quickly. The designed CNN-LSTM algorithm takes the gyro output signal as input, which is connected to a convolution module after passing through the embedding layer; then, the feature vector extracted by the CNN network is reduced in dimension through a maximum pooling layer and is sent to an LSTM module to extract features. Finally, the Sigmoid activation function is used to classify the noise of the gyro output signal to determine whether it is a useful signal. Below, the specific design of each layer in the network is described in detail. The overall structure of the network is shown in Figure 6, and the specific algorithm flow is shown in Figure 6.

When the output of the MEMS gyroscope completes the data preprocessing, the CNN module is first entered. Since the gyro signal vector we process is a one-dimensional vector, one-dimensional convolution is chosen for processing. This step is mainly meant to extract the relevant features in the output of the gyroscope through convolution operation. $x\in R^{d}$ is set to represent the one-dimensional vector with the drift of d corresponding to the i-th time point in the output of the gyroscope, and $x\in R^{l+d}$ represents the input gyro signal, where l represents the length of the output of the gyroscope. Then, the window vector $w_{j}$ of each position j in the signal corresponding to the continuous k-length of the output of the gyroscope can be expressed as:(16)$$w_{j}=[x_{j},x_{j+1},\ldots ,x_{j+k-1}]$$

Then, through the filter in the convolutional layer, the feature vector corresponding to the window vector can be calculated, and the calculation method is shown in Equation (17).
(17)$$c_{j}=f(w_{j}⊙p+b)$$
where ⊙ represents the element-wise multiplication of feature vectors, b∈R is a bias term, and f is a nonlinear mapping function. In the research process, the function is set as the ReLU activation function, which is defined as Equation (18).
(18)f(x)=max(0,x)

During the calculation, if the element in the vector is a regular element, it returns the element; otherwise, it returns to 0. The specific browsing learning process is shown in Algorithm 1.
**Algorithm 1.** CNN model**Input:** x∈Rl+d**Output:** c^**For each position***j***in the signal, perform**|wj=[xj,xj+1,…,xj+k−1]|cj=ReLU(wj⊙p+b)**End**cj=[c1,c2,…,cl+k−1]c^=max(c)//**Max pooling operation**

The feature vector obtained by the convolution layer is the high-dimensional feature of the output signal of the gyro. In order to filter out the redundant noise, only the important features in the output signal of the gyro are retained to avoid the trained network caused by the noise information in the text. In the study, a max pooling layer is added after the CNN module to decrease the dimension of the feature vector, which can also reduce the computational cost of the network. The CNN network has a very good performance in extracting relevant features from the gyro output signal data. However, it cannot correlate current content with past content in conjunction with contextual information in the gyro output signal. Therefore, another deep learning network LSTM is added to the structure to complete the learning of associated features.

The basic structure of a LSTM network consists of a series of repeating units at each time step. In each unit, at time step t, an information storage part ct and three gate functions, namely input gate it, output gate ot and forget gate ft, are used to regulate and manage the information flow of each unit in the LSTM network and to decide how to update the information stored in the current storage unit ct and the current hidden state ht of the unit. The relevant calculation function of each unit in the LSTM network module is listed in Equations (19)–(24).
(19)it=σ(Wi·[ht−1+bi])
(20)ft=σ(Wf·[ht−1,xt]+bf)
(21)qt=tanh(Wq·[ht−1,xt]+bq)
(22)ot=σ(Wo·[ht−1,xt]+bo)
(23)ct=ft⊙ct−1+it⊙qt
(24)ht=ot⊙tanh(ct)
where xt is the input feature in the LSTM network unit, which is extracted by the above CNN module. σ represents the Sigmoid function, ⊙ represents the element-by-element multiplication of feature vectors, W and b represent the weight matrix and offset vector during training, respectively. In the structure of the proposed model, an LSTM module is placed directly after the max pooling layer, containing 64 LSTM units, which use a dropout layer of 0.2 as a regularization parameter to prevent the model from overfitting. The specific training and learning processes are shown in Algorithm 2.
**Algorithm 2**. LSTM mode**Input:** Signal with noise**Output:** Signal**For each time step t, perform**|it=σ(Wi·[ht−1+bi])|ft=σ(Wf·[ht−1,xt]+bf)|qt=tanh(Wq·[ht−1,xt]+bq)|ot=σ(Wo·[ht−1,xt]+bo)|ct=ft⊙ct−1+it⊙qt|ht=ot⊙tanh(ct)**End**

The dense layer is the last layer of the entire model. That is, the fully connected layer in the neural network is used to output the result and to classify the gyro output signal according to the output of the LSTM layer. Since the dataset used for training in this paper divides the gyro output signal into two categories, our classification model is binary. A fully connected layer and sigmoid function are used to provide 0 or 1 predictions for two classes (useful signal and noise), where the sigmoid function is a logical function whose return value is between 0 and 1, as defined in Equation (25).
(25)$$f(x)=\frac{1}{1+e^{-x}}$$

### 3.3. The Algorithm of PSO-SVM

The PSO algorithm, proposed by Kennedy and Eberhardt in 1995, is an intelligent biological algorithm. The algorithm takes advantage of mineral populations in nature and uses their behavior to first solve complex optimization problems with the help of environmental services. Compared with the genetic algorithm, the PSO algorithm has the advantages of speed dissipation, easy implementation, and high precision.

PSO also introduces the thorny problem of how to perform a random search in a D dimensional space with the goal of solving the problem of maximizing or minimizing the objective function, where $X=\left(X_{1},X_{2},\ldots X_{n}\right)$, $\;x_{i}=\left(x_{i1},x_{i2},\ldots ,x_{iD}\right)$, and $v_{i}=\left(v_{i1},v_{i2},\ldots v_{iD}\right)$ represent a population composed of n particles, the position vector of a single particle i, and the velocity vector, respectively. When the particle i searches the D dimensional space, the local optimal solution is the optimal position searched for $pbest_{i}={\left(pbest_{i1},pbest_{i2},\ldots ,pbest_{ig}\right)}^{T}$, and the global optimal solution is the optimal position $\;gbest_{b}={\left(gbest_{b1},gbest_{b2},\ldots ,gbest_{bg}\right)}^{T}$ searched by the entire particle swarm.

In the iterative process, the velocity of each particle is modified and determined according to the position of the local optimal particle, the position of the global optimal particle, and the velocity and position of the particle itself. The local optimal position is the optimal position reached by each particle in the iterative optimization process. The particle velocity calculation and position calculation results are shown in Equations (26) and (27).
(26)$$\;v_{i}(t+1)=wv_{i}\left(t\right)+c_{1}r1(pbest_{i}\left(t\right)-x_{i}\left(t\right))+c_{2}r2(gbest_{i}\left(t\right)-x_{i}\left(t\right))$$
(27)$$x_{i}(t+1)=x_{i}\left(t\right)+v_{i}(t+1)$$
where i=1,2…,N, N is the average number, w is the inertia coefficient, and its value is not negative. When its value is relatively large, the global search ability is strong while the local search ability is weak. When its value is small, the global search is weak, the local search is strong, and the best search results can be obtained. itmax is the maximum number of iterations, $\;w_{ini}$ is the initial inertia weight, the typical weight is $w_{ini}=0.9$. $\;w_{ini}$ is inertia weight when the iteration $w_{end}$ reaches the maximum algebra, and the typical weight is $w_{end}=0.4$. The local and global learning factors are $c_{1},c_{2}$ with the range of $0\leq c_{1},c_{2}\leq 2$, (usually taken as $c_{1}=c_{2}=2$). $r_{1},r_{2}$ are two random numbers in the range (0, 1), where the particle’s position and velocity are limited to [−xmax, xmax],[−vmax, vmax]. $pbest_{i}$ is the local optimal position of the particle, and $gbest_{i}$ is the full optimal position. The basic PSO algorithm steps are summarized below:Step1: For each particle in the population, its fitness needs to be calculated.Step2: For each particle, the best fitness value passed by it is compared with the fitness value. If it is better, it is treated as a locally optimal particle.Step3: For each generation of optimal particles, the global optimal particle is compared with its fitness value, and if it is better, it can be taken as the global optimal particle.Step4: The speed and position of the particles are adjusted according to the above formula.Step5: If the corresponding conditions are not met, go back to step 1.The termination condition of the algorithm iteration is that the optimal position searched by the particle swarm reaches the minimum fitness threshold or the algorithm has iterated to the set maximum number of iterations.

The SVM algorithm is a new machine learning method based on the VC theory of statistical learning and the principle of structural risk minimization. Its core idea is to transform and map the nonlinear data into a high-dimensional linear space to meet the maximum classification distance, so that the classification line can correctly separate the two types of samples, and then, the required optimal classification hyperplane is obtained. As shown in Figure 7, assuming that the given dataset is $\left\{x_{i},y_{i}\right\},i=1,2,\ldots ,N$, $y_{i}\in \left\{-1,+1\right\}$, $x_{i}\in R_{d}$, two triangles and five pointed stars are used to represent two samples on the plane.

The SVM introduces the slack variable $\xi_{i}\geq 0$ and penalty coefficient C(C>0). Then, the solution formula of the hyperplane is:(28)$$\begin{array}{l} \max\sum_{i=1}^{m}\alpha_{i}-\frac{1}{2}\sum_{i=1}^{m}\alpha_{i}\alpha_{j}y_{i}y_{j}(x_{i},x_{j})\\ s.t.0\leq \alpha_{i}\leq C,i=1,\cdot \cdot \cdot ,m\\ \sum_{i=1}^{m}\alpha_{i}y_{i}=0 \end{array}$$

In the formula, $\alpha_{i}$ is the Lagrangian factor solved by the quadratic optimization problem to obtain the optimal classification hyperplane.
(29)$$K(x,x’)=\exp(-\frac{{\left\|x-x’\right\|}^{2}}{\sigma^{2}})$$

Then, the optimal classification hyperplane is:(30)$$f(x)=sign[\sum_{i=1}^{N}\alpha_{i}y_{i}k(x_{i}\cdot x)+b]$$

The algorithm flow of the prediction model establishment based on PSO-SVM is shown in Figure 8. The specific steps of the model establishment are as follows:

Step 1: Obtain relevant data, determine the number of training samples and test the samples, quantify the sample data uniformly, and normalize the quantized data to the interval [0, 1].

Step 2: Set the initial parameters of the PSO-SVM model, including the population size, the number of iterations, the learning factor, the inertia weight, the initial particle position and the initial particle velocity, etc.

Step 3: Evaluate the fitness of each particle and update the individual extremum and global extremum of the particle swarm.

Step 4: Judging whether the end condition is met or the terminating evolutionary algebra is reached, output the optimal penalty factor C and kernel function K; if the judgment result is no, then return to step 3.

Step 5: Input the optimal penalty factor C and kernel function K into the SVM model for sample training, and obtain the globally optimal PSO-SVM model, which is used to predict the data.

### 3.4. The Algorithm of ELM

Given an initial training sample $\left(x_{i},t_{i}\right)$, the input sample is denoted as $x_{i}={\left[x_{i1},x_{i2},\ldots ,x_{in}\right]}^{T}\in R^{n}$, and the output sample is denoted as $t_{i}={\left[t_{i1},t_{i2},\ldots ,t_{im}\right]}^{T}\in R^{m}$. The SLFN network model with hidden layer nodes $L(N_{0}\geq L)$ and activation function is g(x), as shown in Figure 9.

In order to make the learning objective of SLFNs minimize the output, it can be expressed in Equation (31), where $w_{i}={\left[w_{i1},w_{i2},\ldots ,w_{in}\right]}^{T}$ represents the weight vector of the input node and the node connecting the hidden layer, $\;\beta_{i}={\left[\beta_{i1},\beta_{i2},\ldots ,\beta_{im}\right]}^{T}$ represents the weight vector used to connect the output node and the hidden layer node, $b_{i}$ is the bias of the ith hidden layer node, and $w_{i}\cdot x_{j}$ represents the inner product of $w_{i}$ and $x_{j}$.
(31)$$\sum_{i=1}^{L}\beta_{i}g_{i}(x_{i})=\sum_{i=1}^{L}\beta_{i}g(w_{i}\cdot x_{j}+b_{i})=o_{j},j=1,2,\ldots ,N$$
(32)$$\sum_{j=1}^{N}\left\|o_{j}-t_{j}\right\|$$

That is, there is $\beta_{i}$ and $\omega_{i}$ to make Equation (31) true, as shown in Equation (33).
(33)$$\sum_{i=1}^{L}\beta_{i}g(w_{i}\cdot x_{j}+b_{i})=t_{j},j=1,2,\ldots ,N$$

Equation (31) can be simplified to Equations (34) and (35).
(34)Hβ=T
(35)$$H(w_{1},\ldots ,w_{L},b_{1},\ldots b_{L},x_{1},\ldots ,x_{L})={\left[\begin{array}{ccc} g(w_{1}\cdot x_{1}+b_{1}) & \ldots & g(w_{L}\cdot x_{1}+b_{L})\\ \ldots & \ldots & \ldots \\ g(w_{1}\cdot x_{N}+b_{1}) & \ldots & g(w_{L}\cdot x_{N}+b_{L}) \end{array}\right]}_{N\times L}$$

In Equation (35), H is the output matrix of the neural network hidden layer interface, and the column of L is the output matrix H of the i hidden layer node. For some gradient proximity algorithms, when the deployment function is infinite, the input link density and hidden layer irregularity can be randomly generated for training, which will change during training, to ensure that the output link weights β are obtained by the least squares solution in Equation (36).
(36)min‖Hβ-T‖

In the ELM neural network algorithm, random connection weights $w_{i}$ and hidden layer biases $b_{i}$ are obtained, and the output matrix H of the hidden layer is determined. The ELM neural network can be trained to change into a linear system Hβ=T, and the output weights β can be determined, as shown in Equation (37).
(37)$$\hat{\beta }=H^{+}T$$
where $H^{+}$ indicates that the Moore–Penrose generalized inverse is the hidden layer output matrix H, and the norm $\hat{\beta }$ needs to be known as the smallest and most unique. In summary, the steps of the ELM learning method can be summarized as follows:Step1:Given a training set $set(x_{i},t_{i}),(i=1,2,\ldots ,N)$, the activation function is g(x), the number of hidden layer nodes is L, the random initialization input weight is $w_{i}$, and the hidden layer node offset is $b_{i}$.Step2: Calculate the hidden layer output matrix H.Step3: Calculate the output weight matrix β.

Due to the ELM algorithm, the output weight matrix can be obtained only by calculation, which can greatly reduce the training complexity and greatly improve the training efficiency. The core idea of the ELM algorithm is to calculate the output weight matrix of the hidden layer through the Moore–Penrose generalized inverse and to transform the training process of the traditional SLFN model into a least-squares solution problem, and the calculated solution is unique. The difference between ELM and traditional SLFNs is that ELM does not need to iteratively adjust all node parameters of the hidden layer during model building and training processes. Meanwhile, the node parameters are randomly generated during the network training process. Furthermore, when training the network model, there is no need to adjust the parameters.

## 4. The Improved VMD and ELM Algorithm Based on CNN-LSTM and PSO-SVM

In this paper, an improved VMD and ELM algorithm is designed and proposed based on CNN-LSTM and PSO-SVM. This optimization algorithm identifies the noise of different frequencies in the gyro output signal through VMD and then passes the high-, medium- and low-frequency noise through the CNN-LSTM. PSO-SVM algorithms are used for modeling analysis, and then, the optimized noise signal is passed through the ELM neural network to establish a temperature compensation model, and finally, the optimized gyro output signal is obtained. Figure 10 describes the specific process of this improved method:

(1) Preprocess the original signal output by the MEMS gyroscope, and then, perform frequency division processing on the preprocessed signal through the VMD algorithm to obtain three groups of signals, which are high-frequency noise signal $\eta_{high}(t)$, intermediate-frequency temperature noise signal $\eta_{medium}(t)$ and low-frequency noise signal $\eta_{low}(t)$.

(2) The high-frequency noise signal $\eta_{high}(t)$ is discarded, and then, the low-frequency noise signal $\eta_{low}(t)$ is modeled through the CNN-LSTM algorithm to establish the high-frequency signal compensation model and to obtain the modeled signal $\eta_{low}^{1}(t)$.

(3) The intermediate-frequency temperature noise compensation model is established by PSO-SVM, the intermediate-frequency temperature noise signal $\eta_{medium}(t)$ is compensated by the PSO-SVM algorithm, and the compensated signal $\eta_{medium}^{2}(t)$ is obtained.

(4) The temperature T, temperature change rate dT, low-frequency noise compensation signal $\eta_{low}^{1}(t)$, and intermediate-frequency temperature noise compensation signal $\eta_{medium}^{2}(t)$ are used as the input layer of the ELM neural network, and a temperature compensation analysis model is established. Finally, the final temperature compensation output of the gyroscope is obtained at the output layer of the ELM neural network signal.

## 5. Experiment and Analysis

### 5.1. Experiment Process

The MEMS is kept on the temperature-controlled oven in order to output the signal of the MEMS vibration gyroscope, which is not influenced by outside temperature variation. Then, the ranges of temperature and temperature rate are set from −40 to 60 °C and 1 °C/min, respectively. Firstly, the initial temperature should be set as 60 °C and kept for an hour in order to ensure that the inside temperature of the MEMS vibration gyroscope is stable at 60 °C. Secondly, the temperature is reduced at the speed of −1 °C/min into −40 °C, and the temperature is kept for an hour to make sure the inside temperature of the MEMS vibration gyroscope is −40 °C. The temperature-controlled oven should stay at 10 °C for an hour in order to collect the output of the MEMS vibration gyroscope and to ensure that the inside temperature of the MEMS vibration gyroscope is stable and equal to the oven temperature. The equipment of the temperature test and the process of the temperature experiment are shown in Figure 11 and Figure 12 [29,30,31].

Based on the designed full-temperature experiment, data were collected at fixed points, the test was repeated for 10 days, ten sets of data were collected, and the sixth set of data with the average value of Allan variation through 10 sets of data was used for analysis. The output data from −40 to 60 °C and from 60 to −40 °C are shown in Figure 13 and Figure 14, respectively.

### 5.2. Data Analysis

It can be seen from the decomposition diagram that the output signal consists of a low-frequency noise signal, intermediate-frequency noise signal, and high-frequency noise signal. The low-frequency noise signal, intermediate-frequency noise signal, and high-frequency noise are extracted through VMD decomposition, and eight natural mode functions are obtained (IMF1–IMF8), which represent different characteristics of different noise signals in the output signal. If each IMF is processed, the calculation consumes a lot, and it is easy to destroy the static information, resulting in errors; thus, this entropy is used for the modal function. Figure 15a shows the IMF1–IMF8 based on VMD from −40 to 60 °C, and Figure 15b shows the IMF1-IMF8 based on VMD from 60 to −40 °C. According to the sequence autocorrelation and complexity, the IMF is divided into high-frequency noise signal $\eta_{high}(t)$, intermediate-frequency temperature noise signal $\eta_{medium}(t)$, and low-frequency noise signal $\eta_{low}(t)$.

If the sample entropy value of the IMF is greater than 0.6, it is a pure noise term that does not contain any useful signals, and it can be deleted directly. If the sample entropy value of the IMF is between 0.3 and 0.5, it is intermediate-frequency temperature noise, which also includes effective temperature characteristics and noise signals. The PSO-SVM method is used to deal with this part of the noise, because it is the most important part of the temperature compensation model. If the signal sample entropy value is between 0.2 and 0.3, the signal is a low-frequency noise signal, using CNN-LSTM to build the compensation model. The specific classification of high-frequency noise signal $\eta_{high}(t)$, intermediate-frequency temperature noise signal $\eta_{medium}(t)$ and low-frequency noise signal $\eta_{low}(t)$ is shown in Figure 16.

The IMF2 and IMF3 sequences decomposed by VMD are selected as the training and test datasets of CNN-LSTM, and the operating parameters of CNN-LSTM are set. The specific parameters are shown in Table 1. After repeated experiments and network optimization, the RMSE error value and prediction data are shown in Figure 17 and Figure 18. Among them, Figure 17 shows the low-frequency noise filtering and optimization during the heating process (−40–60 °C), and Figure 18 shows the low-frequency noise filtering and optimization during the cooling process (60–−40 °C).

SVM is used to train temperature noise data and to improve the relatively better C and γ of the sexual particle group optimization algorithm of SVM. The range of C and γ are set from 0 to 10, the value of the inertial factor W is 0.5, the values of the learning factors $C_{1}$ and $C_{2}$ are set to 1.46, the total number of particles is set to 100, and the number of iterations is set to 50. The initial state and the end of the iteration of the searches of particle group optimization algorithms are shown in Figure 19. It can be seen from the figure that in the initial state, PSO randomly generates several particles. Each particle represents a set of C and γ values, and the relatively better parameters are finalized by continuous iteration updates. Ten tests are conducted under the setting conditions, and then, the average value is taken as the final result. The relatively better parameter value C under the setting conditions is 7.20, and γ is 6.75. The support vector machine model in this parameter is relatively better. Figure 20 shows the temperature noise compensation based on PSO-SVM. Among them, the compensation can remove the effects of temperature noise on the output signal of the PSO-SVM with good compensation effects.

Low-frequency noise after CNN-LSTM treatment, temperature noise after PSO-SVM treatment, and normal value noise are reconstructed through the ELM algorithm. The input signal is the low-frequency noise after CNN-LSTM treatment, the temperature noise after the PSO-SVM treatment, constant noise, temperature, and temperature change rate. The reconstructed output signal is shown in Figure 21 and Figure 22. It can be concluded that the method used in this study can dramatically reduce the impact of temperature on the performance of the gyroscope and that it greatly improves the output accuracy of the gyroscope.

Finally, the prominent feature of Allan variance is that it can be easy to represent and identify various sources of error and the contribution of the whole noise statistical characteristics as it has the advantages of easy calculation and easy separation. The Allan variance is widely used in gyroscope performance analysis as an IEEE-approved standard analysis method. The results are shown in Table 2. On the one side, by using the improved method, the outputs of the MEMS gyroscope that ranged from −40 to 60 °C were reduced, and the temperature drift was greatly reduced. For example, the factor of Q was reduced from 1.2419 × 10^−4^ to 1.0533 × 10^−6^, the factor of B was reduced from 0.0087 to 1.8772 × 10^−4^, the factor of N was reduced from 2.0978 × 10^−5^ to 1.4985 × 10^−6^. On the other side, the outputs of the MEMS gyroscope that ranged from 60 to −40 °C were reduced. The factor of Q was reduced from 2.9808 × 10^−4^ to 2.4430 × 10^−6^, the factor of B was reduced from 0.0145 to 7.2426 × 10^−4^, and the factor of N was reduced from 4.5072 × 10^−5^ to 1.0523 × 10^−5^.

## 6. Conclusions

The detailed temperature error of the MEMS gyroscope was studied by proposing an improved VMD-ELM algorithm based on CNN-LSTM and PSO-SVM. Within the improved fusion method based on the temperature experiment and compared experiment, the output of the MEMS gyroscope went through a process of searching for temperature error, establishing a temperature error compensation model, and filtering. The main findings are as follows:

(1) The improved VMD-ELM algorithm based on CNN-LSTM and PSO-SVM combined VMD, CNN-LSTM, PSO-SVM, and ELM. The final output of the MEMS gyroscope greatly decreased compared to that of the Allan variance method, which indicated good feasibility and effectiveness of the algorithms based on the novel method.

(2) Using the improved method, the output of the MEMS gyroscope ranging from −40 to 60 °C reduced, and the temperature drift dramatically declined. For example, the factor of Q reduced from 1.2419 × 10^−4^ to 1.0533 × 10^−6^, the factor of B reduced from 0.0087 to 1.8772 × 10^−4^, and the factor of N reduced from 2.0978 × 10^−5^ to 1.4985 × 10^−6^. Furthermore, the output of the MEMS gyroscope ranging from 60 to −40 °C reduced. The factor of Q reduced from 2.9808 × 10^−4^ to 2.4430 × 10^−6^, the factor of B reduced from 0.0145 to 7.2426 × 10^−4^, and the factor of N reduced from 4.5072 × 10^−5^ to 1.0523 × 10^−5^.

(3) The experiments show that the method proposed in this study can greatly compensate the gyro output signal to obtain zero bias stability, zero bias instability, and angle random walking with a stable effect. The experimental results show that the compensation of the novel VMD-ELM algorithm based on CNN-LSTM and PSO-SVM significantly improved with obvious compensation effects to provide certain engineering application value.

## Figures and Tables

**Figure 1 micromachines-13-02056-f001:**
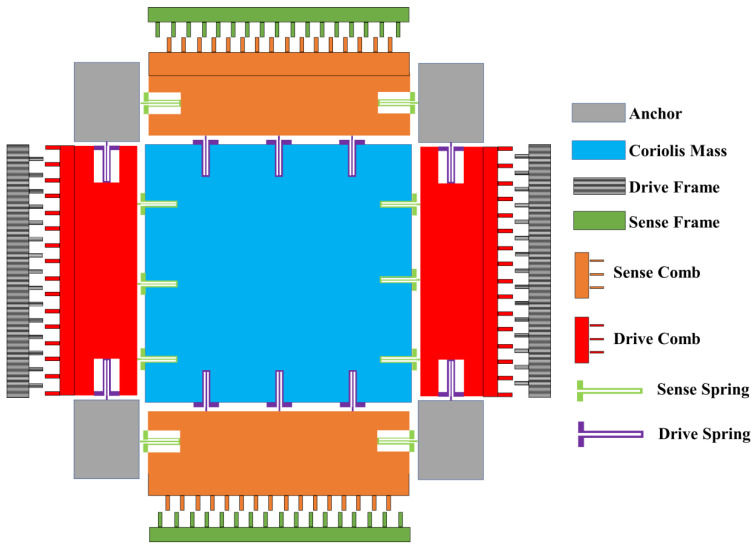
The structure of the MEMS gyroscope.

**Figure 2 micromachines-13-02056-f002:**
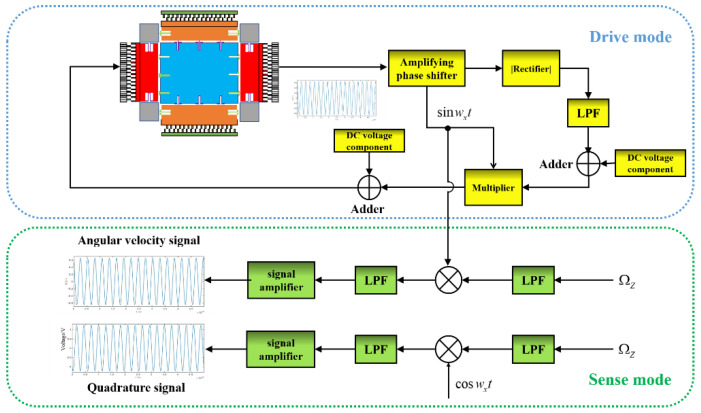
Circuit structure of MEMS gyroscope.

**Figure 3 micromachines-13-02056-f003:**
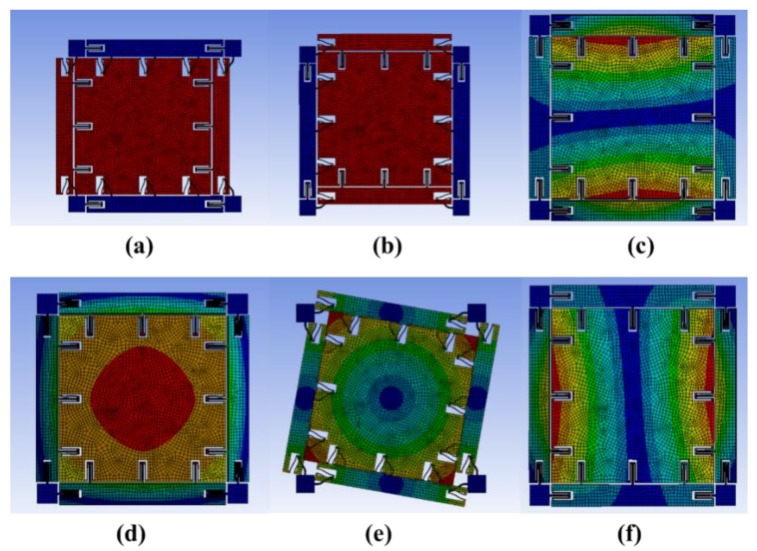
The vibration mode of MEMS gyroscope: (**a**) drive mode; (**b**) sense mode; (**c**) irrelevant mode; (**d**) irrelevant mod; (**e**) irrelevant mode; (**f**) irrelevant mode.

**Figure 4 micromachines-13-02056-f004:**
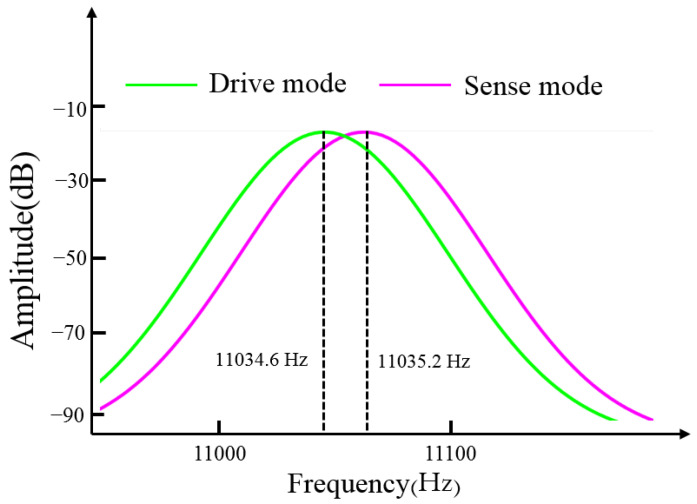
Frequency difference between driving mode and sensitive mode.

**Figure 5 micromachines-13-02056-f005:**
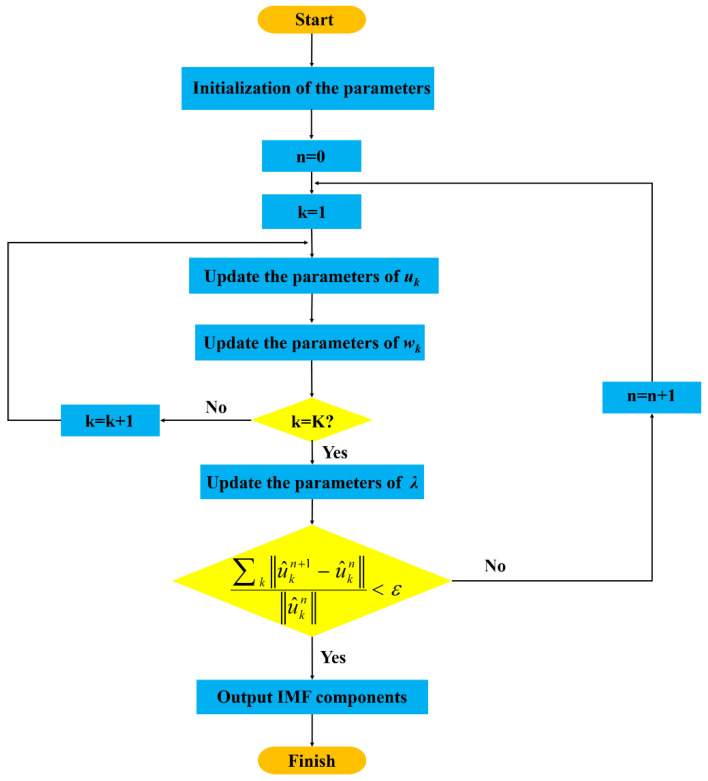
VMD algorithm flow chart.

**Figure 6 micromachines-13-02056-f006:**
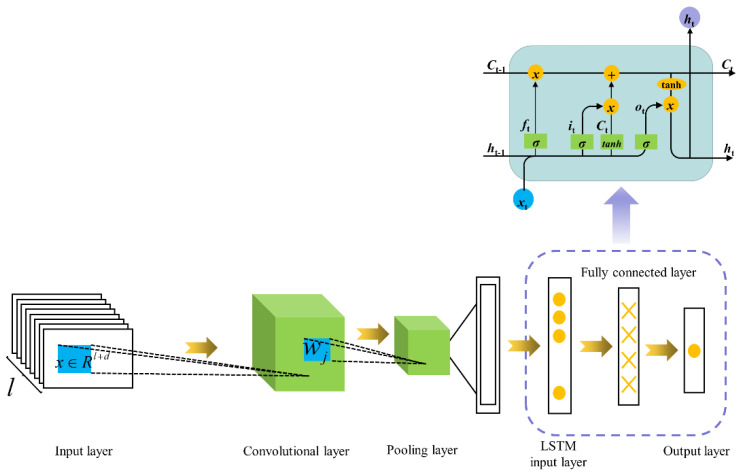
CNN-LSTM algorithm flow chart.

**Figure 7 micromachines-13-02056-f007:**
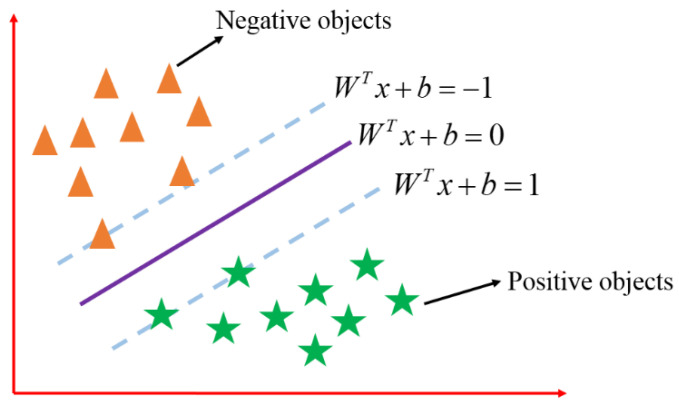
Schematic diagram of support vector machine.

**Figure 8 micromachines-13-02056-f008:**
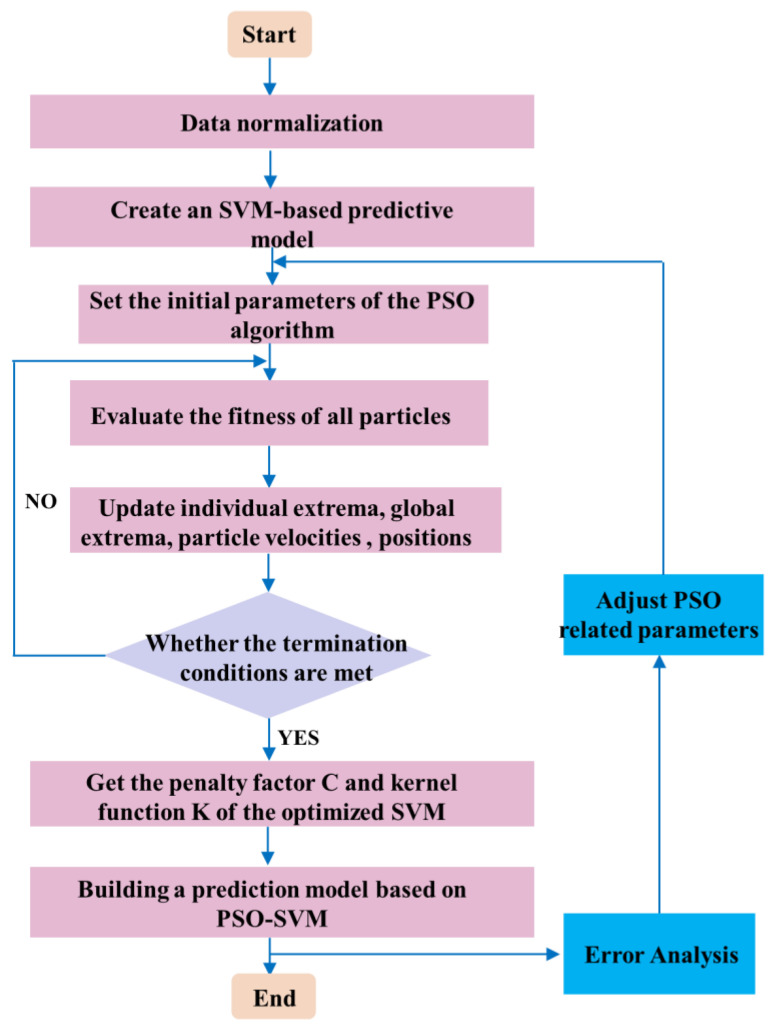
PSO-SVM algorithm flow chart.

**Figure 9 micromachines-13-02056-f009:**
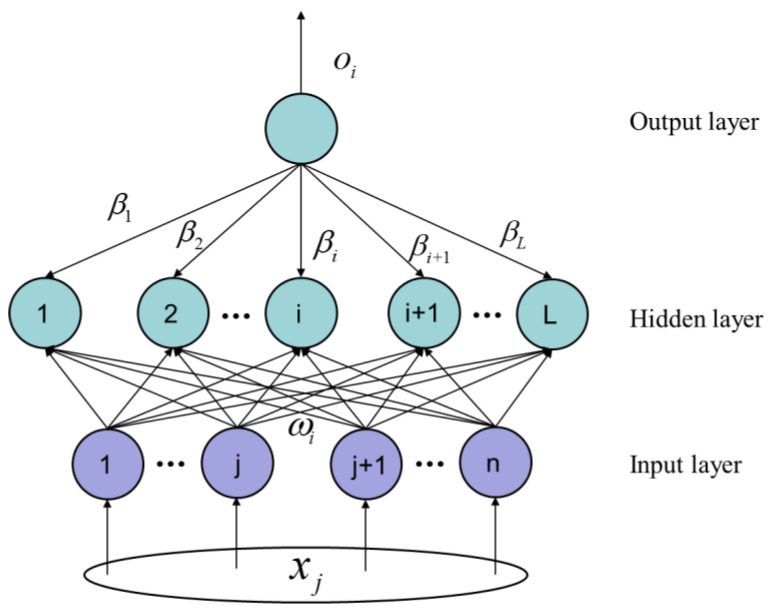
SLFN network model.

**Figure 10 micromachines-13-02056-f010:**
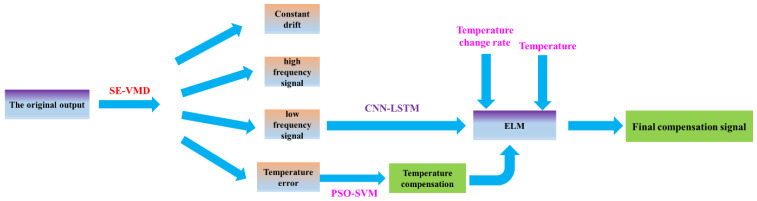
The specific process of the improved method.

**Figure 11 micromachines-13-02056-f011:**
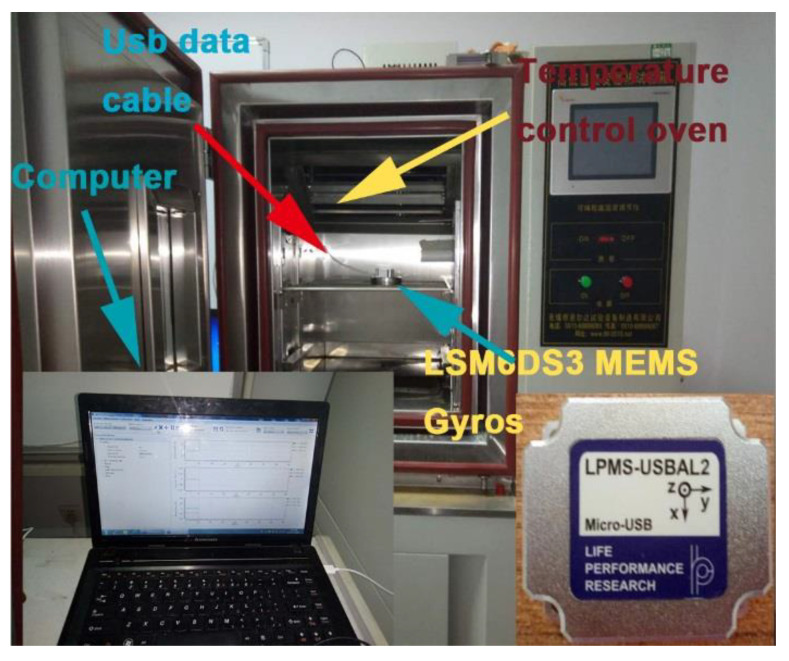
Equipment of the temperature test.

**Figure 12 micromachines-13-02056-f012:**
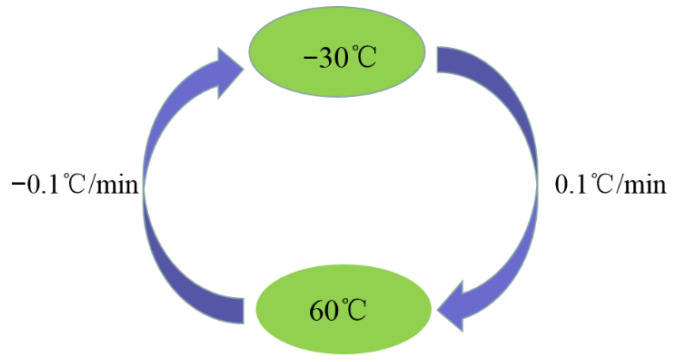
Process of the full relative humidity experiment.

**Figure 13 micromachines-13-02056-f013:**
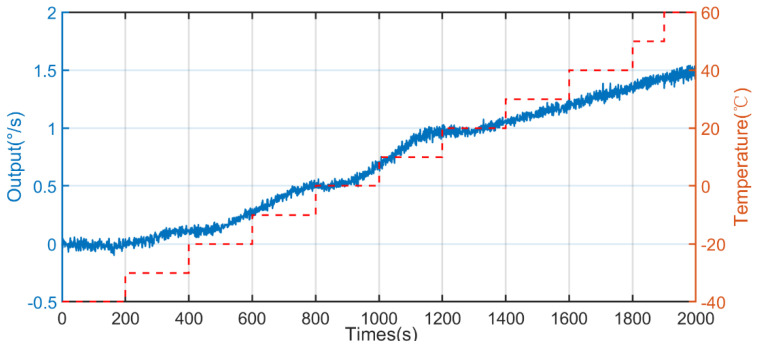
The output of the gyroscope based on the X axis from −40 to 60 °C.

**Figure 14 micromachines-13-02056-f014:**
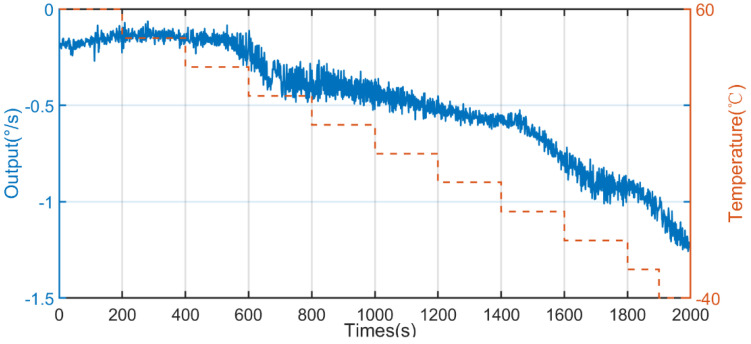
The output of the gyroscope based on the X axis from 60 to −40 °C.

**Figure 15 micromachines-13-02056-f015:**
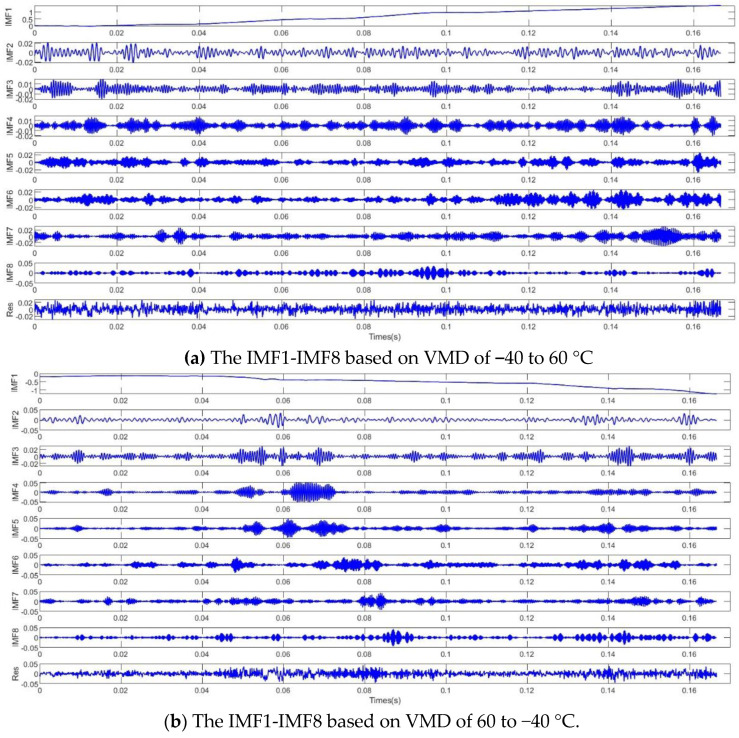
The IMF1-IMF8 based on VMD of (**a**) −40 to 60 °C and (**b**) 60 to −40 °C.

**Figure 16 micromachines-13-02056-f016:**
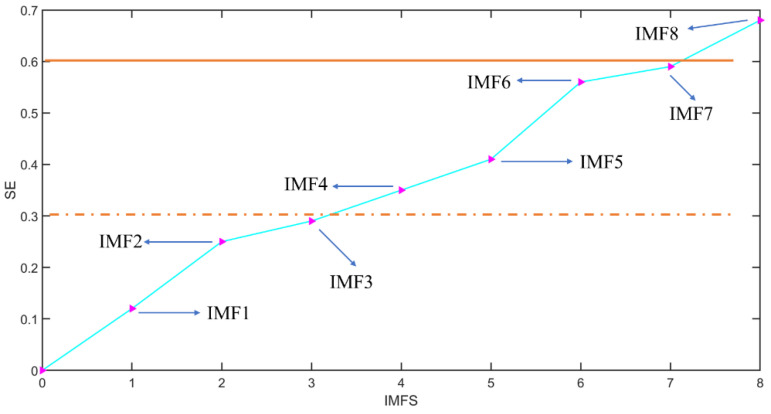
Noise classification based on SE-VMD and CNN-LSTM.

**Figure 17 micromachines-13-02056-f017:**
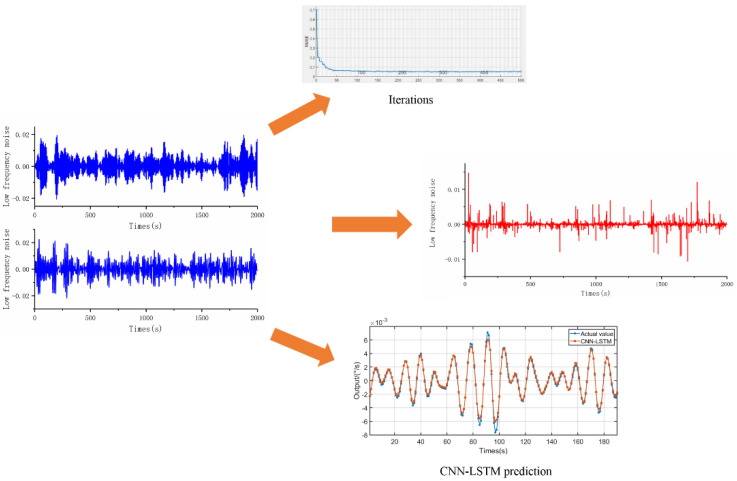
Noise denoising based on CNN-LSTM (−40 °C−60 °C).

**Figure 18 micromachines-13-02056-f018:**
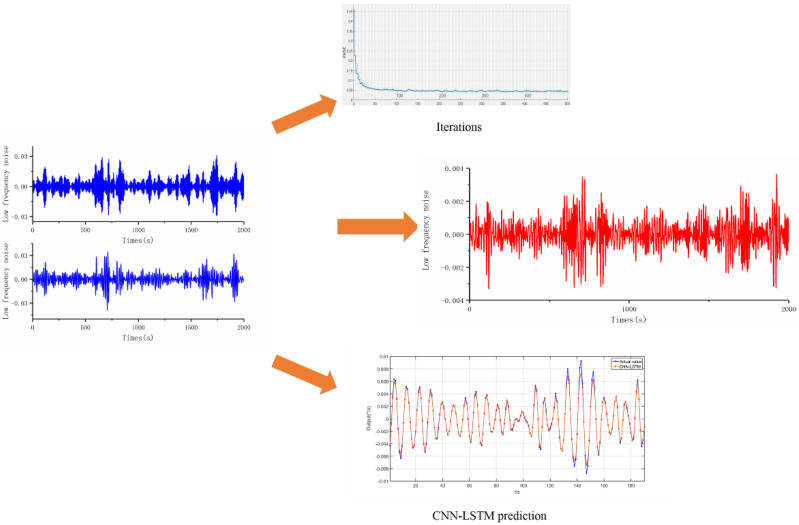
Noise denoising based on CNN-LSTM (60 °C–−40 °C).

**Figure 19 micromachines-13-02056-f019:**
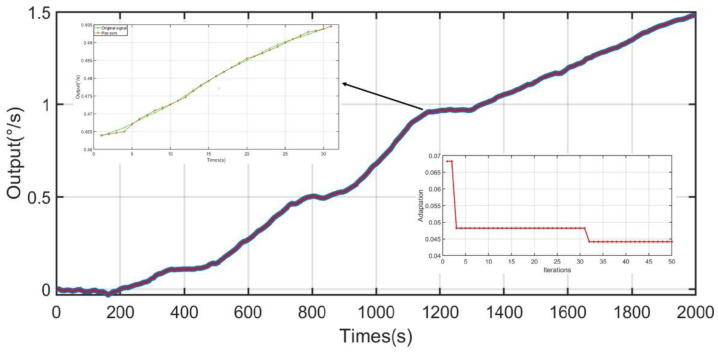
Temperature noise denoising based on PSO-SVM (−40–60 °C).

**Figure 20 micromachines-13-02056-f020:**
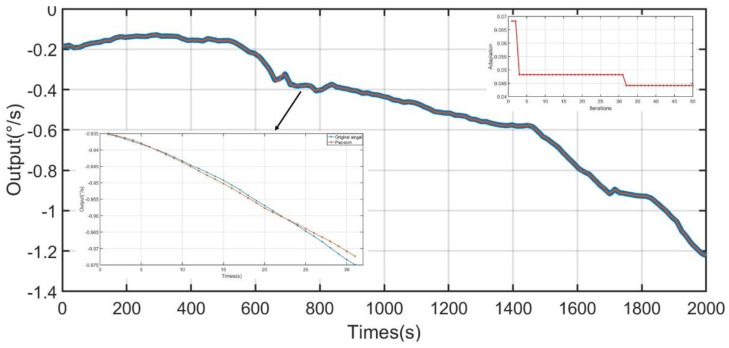
Temperature noise denoising based on PSO-SVM (60–−40 °C).

**Figure 21 micromachines-13-02056-f021:**
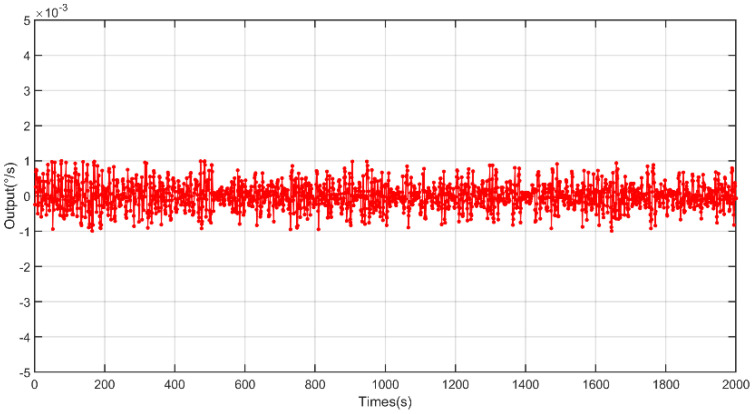
Signal reconstructed based on ELM (−40 °C−60 °C).

**Figure 22 micromachines-13-02056-f022:**
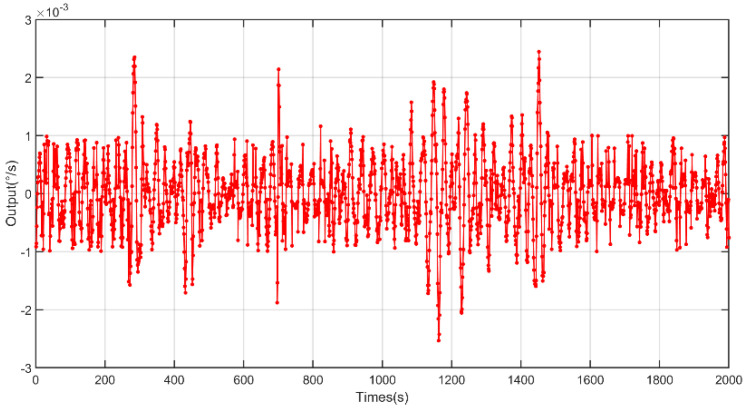
Signal reconstructed based on ELM (60 °C–−40 °C).

**Table 1 micromachines-13-02056-t001:** CNN-LSTM specific operating parameters.

Parameter	Specific Operation
Number of CNN-LSTM modules	4
Input	Noise
Data normalization	Min–Max
Number of LSTM units	5
Number of LSTM layers	2
Softmax layer	Softmax
Loss function	MSE
Number of iteration rounds	500

**Table 2 micromachines-13-02056-t002:** The Allan variance of the compared experiment based on original signal and the improved method.

	Original Signal		Denoising Signal	
Allan	−40–60 °C	60—40 °C	−40–60 °C	60—40 °C
*Q* (${}^{\circ }$)	1.2419 × 10^−4^	2.9808 × 10^−4^	1.0533 × 10^−6^	2.4430 × 10^−6^
*N* (${}^{\circ }/\sqrt{h}$)	2.0978 × 10^−5^	4.5072 × 10^−5^	1.4985 × 10^−6^	1.0523 × 10^−5^
*B* (${}^{\circ }/h$)	0.0087	0.0145	1.8772 × 10^−4^	7.2426 × 10^−4^

## Data Availability

Not applicable.

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
