# Peer review of "Improved VMD-ELM Algorithm for MEMS Gyroscope of Temperature Compensation Model Based on CNN-LSTM and PSO-SVM"

_micromachines, 2022, doi:10.3390/mi13122056_

Round 1
Reviewer 1 Report
The paper considers the possibility of improving the accuracy of MEMS gyroscopes in the temperature range of -40 ... +60 C through the use of an improved information processing method. The presented experimental data confirm the correctness of the conclusions drawn. The paper may be useful to specialists working in the field of development and production of MEMS gyroscopes.
There are some remarks.
1. The authors investigated the operation of the microgyroscope and demonstrated that the use of the proposed data processing method reduced the factors Q, B, N. Considering that the authors examined one microgyroscope, the following phrase in the abstract and conclusion (paragraph 6(2)) should be supplemented with the following words:
“For example, in the studied specimen of the MEMS gyroscope the factor of Q reduced…..”
In addition, the abstract should indicate that the factors Q, B, N refer to the Allan variation.
2. In the list of references for each cited paper, the full name of only the first author is indicated. I believe that for each cited paper, you should indicate the full names of all authors, as well as the DOI, and carefully check all references. For example, [15] does not specify the Article ID, and [7] does not specify page numbers.
3. On the page 2 of the manuscript, the paper [7] is incorrectly quoted: “For example, Tamara G [7] presents …”
Correct: “For example, Nesterenko [7] presents …” (as Nesterenko is a family name).
4. On the pages 2 and 3, the abbreviation MEMS is introduced three times, but even earlier (on the page 1) this abbreviation is already used. Of course, this abbreviation is in general use, but if the authors introduce it, then this must be done once at the first use.
5. On the page 5 in paragraph 1 “m” is probably missing. It should probably be: “Where m is the main mass…”
6. Figure 2 shows the extraction of the angular velocity signal and the quadrature signal, but both circuits are the same. Probably in the quadrature channel there should be an additional phase shift of the reference signal.
7. On the page 7, in the last sentence of paragraph 1, there is the phrase “in the coming year”. It is not clear what the authors had in mind. It probably just needs to be removed.
8. On the page 22 (paragraph 2), the authors incorrectly refer to Table 4. Correct: Table 5.
9. On the page 25 (paragraph 1), the authors incorrectly refer to Table 5. Correct: Table 6.
After correcting these remarks, this paper may be published in Micromachines journal.
Author Response
Dear Expert:
Thank you so much for your professinoal suggestions, by followed which, we revised the paper, and we attached the reply. Please do not hesitate to contact us if there is something we can help!
Bests,
Huiliang Cao

Reviewer 2 Report
1. An extensive English editing is required.
2. The introduction is more like a short, however comprehensive, overview of different temperature compensation techniques, most of which are neither critisized, nor utilized in the research. Is it really needed though?
Units in lines 8-10 on p.3 are not clear for me. So as in line 29 (1.667 Ëš/ph...)
Abbreviations used in the final paragraph of the introduction (CNN-LSTM, PSO-SVM) are used, but not intriducted. So as some others further up.
3. Next the authors talk about the structure of a MEMS Gyro. The one presented is a typical one, but not a single one existing. However, it is not clear if the method depend on the exact type/construction of gyro or it is universal? If it is universal, then do we really need this part?
In the final part of this paragraph we see results of a FEM simulation. Was it performed by the authors? I do not see that its results are used somehow further in the research. If the are, then it would be better to add some more details about the model itself, or address to some other paper containing them.
Figure 4, while it almost has no numbers, seems useless, as it does not give any more information, then a value of 2 frequencies.
4. The algorithm in Fig. 5. What I see is that we put k=1, do nothing with k and then ask if it is still equal to 1. Furthermore, if it is not, we increment k by 1 and go bcak to instruction "k=1". Seems that something is wrong there.
5. Several times the authors use the term "length of output". What is it?
6. Describing SVM method autors address to V.N. Vapnik, but do not make any citation. Also, describing Fig. 7 authors say "two circles and forks are used to represent two samples on the plane". However, I do not see any forks or circles in Fig. 7.
7. At page 18 (out of 27) we finally come to the method proposed by the authors. Meaning that everyhing before that was kinf of an overview of existing methods. Is it necessary to give them in so much details?
8. Experiment process part. Here authors say "the 6th set of data with better data was used for analysis". How do you estimate the quality of data? And what happened to other samples? Is your algorithm working only once in 10 experiments? Also, Fig. 12 seems useless for me as well.
Fig. 13 should show the output signal for different temperatures. And we know that at 60 degrees Celsium the gyro was held for an hour, but this data is not on the plot. Meaning we see no data for the case of +60 degrees.
The same is for Fig. 14. We do not see any data for the last period (-40 degrees).
9. The test performed by the authors was just heating and cooling and the algorithm succesfully compensated this environment changes. What happends when there would be other influences? For example linear acceleration, or magnetic fields? And, the most important question is what will happen to the signal from the rotation speed? Is it passing through the algorithm safely in aby case or there are some limitations (on the rotation speed frequency, for example)?
Author Response

(The authors gave the same response as above.)

Round 2
Reviewer 2 Report
1. Fig. 5. Some improvements are made but I still see an endless loop since after k=k+1 we have k=1 every time.

Author Response
Dear Expert:
Thank you so much for your time and the positive comment for our work, your conclusion and suggestions are quite accurate, constructive and professional.
Now, we would like to answer your comments and questions one by one in the attached file.

Round 3
Reviewer 2 Report
Thank you very much for your answers and improvements. I find this article much clearer now.